# Are Interactions between Epicardial Adipose Tissue, Cardiac Fibroblasts and Cardiac Myocytes Instrumental in Atrial Fibrosis and Atrial Fibrillation?

**DOI:** 10.3390/cells10092501

**Published:** 2021-09-21

**Authors:** Anirudh Krishnan, Emily Chilton, Jaishankar Raman, Pankaj Saxena, Craig McFarlane, Alexandra F. Trollope, Robert Kinobe, Lisa Chilton

**Affiliations:** 1College of Medicine and Dentistry, James Cook University, Townsville, QLD 4811, Australia; Anirudh.krishnan@jcu.edu.au; 2Department of Biomedical Physiology and Kinesiology, Simon Fraser University, Burnaby, BC V5A 1S6, Canada; emily_chilton@sfu.ca; 3Austin & St Vincent’s Hospitals, Melbourne University, Melbourne, VIC 3010, Australia; jraman@unimelb.edu.au; 4Applied Artificial Intelligence Institute, Deakin University, Melbourne, VIC 3217, Australia; 5Department of Surgery, Oregon Health and Science University, Portland, OR 97239, USA; 6School of Engineering, University of Illinois at Urbana-Champaign, Champaign, IL 61820, USA; 7Department of Cardiothoracic Surgery, Townsville University Hospital, Townsville, QLD 4814, Australia; pankaj.saxena@jcu.edu.au; 8Centre for Tropical Bioinformatics and Molecular Biology, Australian Institute of Tropical Health and Medicine, College of Public Health, Medical and Veterinary Sciences, James Cook University, Townsville, QLD 4811, Australia; craig.mcfarlane@jcu.edu.au; 9Centre for Molecular Therapeutics, Australian Institute of Tropical Health and Medicine, College of Medicine and Dentistry, James Cook University, Townsville, QLD 4811, Australia; alexandra.trollope@jcu.edu.au; 10Centre for Molecular Therapeutics, Australian Institute of Tropical Health and Medicine, College of Public Health, Medical and Veterinary Sciences, James Cook University, Townsville, QLD 4811, Australia; robert.kinobe@jcu.edu.au

**Keywords:** myofibroblast, adipokine, electrotonic coupling, current sink

## Abstract

Atrial fibrillation is very common among the elderly and/or obese. While myocardial fibrosis is associated with atrial fibrillation, the exact mechanisms within atrial myocytes and surrounding non-myocytes are not fully understood. This review considers the potential roles of myocardial fibroblasts and myofibroblasts in fibrosis and modulating myocyte electrophysiology through electrotonic interactions. Coupling with (myo)fibroblasts in vitro and in silico prolonged myocyte action potential duration and caused resting depolarization; an optogenetic study has verified in vivo that fibroblasts depolarized when coupled myocytes produced action potentials. This review also introduces another non-myocyte which may modulate both myocardial (myo)fibroblasts and myocytes: epicardial adipose tissue. Epicardial adipocytes are in intimate contact with myocytes and (myo)fibroblasts and may infiltrate the myocardium. Adipocytes secrete numerous adipokines which modulate (myo)fibroblast and myocyte physiology. These adipokines are protective in healthy hearts, preventing inflammation and fibrosis. However, adipokines secreted from adipocytes may switch to pro-inflammatory and pro-fibrotic, associated with reactive oxygen species generation. Pro-fibrotic adipokines stimulate myofibroblast differentiation, causing pronounced fibrosis in the epicardial adipose tissue and the myocardium. Adipose tissue also influences myocyte electrophysiology, via the adipokines and/or through electrotonic interactions. Deeper understanding of the interactions between myocytes and non-myocytes is important to understand and manage atrial fibrillation.

## 1. Introduction

Atrial fibrillation is characterized by chaotic, ectopic electrical activity and uncoordinated contraction [1]. It is strongly associated with an increased risk and severity of emboli or strokes [2,3]. Atrial fibrillation is also associated with significantly greater risk of heart failure and higher overall mortality [3].

The 2017 Heart Rhythm Society Consensus Statement defines nine types of atrial fibrillation, including episodic, paroxysmal and persistent forms [4]; the various expressions and durations of atrial fibrillation may reflect progressive remodeling and severity. However, different forms of atrial fibrillation may also reflect distinct pathophysiological changes. Where possible, the type of atrial fibrillation suffered by patients within the reviewed literature will be identified.

Increasing age [5], inflammation [1] and obesity [6] are all correlated with increased risk of persistent atrial fibrillation. The most important and frequent cause of atrial fibrillation is atrial dilatation with accompanying atrial fibrosis and remodeling [1]. Each of these conditions may be associated with structural and electrical atrial remodeling, with fibrosis as an important step in initiation and progression of atrial fibrillation [7]. The cardiac extracellular matrix is maintained primarily by fibroblasts [8]; activation of fibroblasts to myofibroblasts is associated with fibrosis and development of atrial fibrillation [7]. There are complex and interesting electrical, structural and chemical interactions [9,10,11,12] between cardiac fibroblasts and myofibroblasts (collectively: (myo)fibroblasts) and myocytes, which are strongly affected by pathophysiological changes which give rise to inflammation, ischemia, fibrosis and altered morphology [11]. These interactions may be instrumental in atrial remodeling which generates a substrate for fibrillation.

The association of atrial fibrillation with obesity opens a new avenue of investigation. Adipose tissue present on the epicardial surface is in direct contact with the myocytes and fibroblasts within the myocardium. Adipocytes, fibroblasts and myocytes are known to be in chemical communication. Epicardial adipose tissue also infiltrates the myocardium, influencing the architecture [13]. Such fatty infiltrates may disrupt the normal pattern of action potentials within the myocardium, slowing conduction and creating electrical heterogeneity which promotes dysrhythmias [13]. Adipocyte dysfunction may be an important contributing factor in the complex pathogenesis of atrial fibrillation [13,14].

This review will consider the role non-myocytes play in modulating the extracellular matrix and myocytes in the pathophysiology underpinning atrial fibrillation. Interactions among myocytes, (myo)fibroblasts and epicardial adipose tissue will be explored, studying how the physical, chemical and electrotonic interplay may lead to fibrosis and both trigger and perpetuate atrial fibrillation.

## 2. Roles of Fibroblasts and Myofibroblasts in Fibrosis and Fibrillation

Under healthy conditions, the myocardium is protected from developing re-entry loops which would give rise to fibrillation. However, with aging, the amount of fibrosis increases [8,15]. Inflammation and coronary artery disease are also more common and more severe in the elderly [15]. Both coronary artery and ischemic heart disease are associated with dysrhythmias, in part due to ischemic conditions, while inflammation is strongly associated with fibrosis and dysrhythmias. Inflammatory conditions associated with rheumatic heart disease and volume overload due to mitral regurgitation also predispose atria to dilate, followed by a propensity for atrial fibrillation [1].

Cardiac muscle exists as an electrical syncytium, in which the sequence of muscle activation optimizes efficient ejection while minimizing the risk of re-entry loop formation. The orderly sequence of action potentials across the myocardium is determined by the architecture of the myocardium and the biophysical properties of myocytes [16]. The extracellular matrix provides a scaffold for myocyte orientation, which are encased individually by the endomysium and organized into ropes by the thicker perimysium [17,18]. Cardiac fibroblasts are primarily responsible for maintaining the extracellular matrix of the heart, adapting as required to mechanical strain across the chambers. Chemical communication between fibroblasts and myocytes appears to influence both cell types. A growing body of evidence from in vivo optogenetic membrane potential recordings from myocytes and (myo)fibroblasts in situ in the myocardium [19], in vitro studies [10,20,21,22] and in silico modeling [23,24,25,26] suggest that (myo)fibroblasts are part of the electrical syncytium, influencing myocyte electrophysiology.

### 2.1. Reparative and Reactive Fibrosis

While fibroblasts are the most populous cell type in the heart, the relative volume of fibroblasts is very small when compared to myocytes, particularly in the ventricles [27,28]. The relative disparity in volume of myocytes to fibroblasts appears less pronounced in the atria, where the muscle mass is reduced. In a detailed study investigating the cellular components and architecture of rabbit SA node, bands of myocytes were surrounded by sheets of fibroblasts, running parallel with the myocytes within substantial perimysial bands [16].

Wound healing is associated with activation of fibroblasts to a ‘super-charged’ phenotype, myofibroblasts [17,29,30]. Like fibroblasts, myofibroblasts are capable of dividing, of migrating to areas of inflammation and of secreting extracellular matrix proteins and signaling factors [31,32]. Myofibroblasts are capable of secreting collagen at a much more rapid rate and secrete a different suite of autocrine/paracrine signaling molecules from fibroblasts. In addition, myofibroblasts are capable of contraction, important for scar contraction during wound-healing [31,32].

Fibrosis within the heart may occur in many forms [33], including reparative and reactive responses [31,32,33,34] which may occur simultaneously within the heart. In reparative (aka replacement) fibrosis, myofibroblasts respond to local signals to assist with wound healing where cardiac myocytes have been lost to infarction or other causes. This type of fibrosis results in a scar through a similar granulation process as seen in wound healing in other organs. In contrast, reactive fibrosis occurs in the absence of necrosis. It may occur in heart failure where the renin-angiotensin-aldosterone system is upregulated and/or the heart wall is suffering from hemodynamic overload [31,32,34].

Reactive fibrosis is primarily interstitial, surrounding myocytes in increasing amounts of perimysial collagen [34]. Endomysial connective tissue encasing each myocyte is also increased in reactive fibrosis (Figure 1). Increased perimysial and endomysial connective tissue increases the diffusion distance from the vasculature, predisposing the myocytes and fibroblasts to ischemia and dysrhythmias. Both reparative and reactive fibrosis may disrupt the electrical syncytium, also predisposing the heart to dysrhythmias [34]. Indeed, myocardial fibrosis is associated with reduced side-to-side conduction of depolarization [35], slowed conduction velocity and ‘zigzag’ course of activation [36].

### 2.2. Homo- and Heterocellular Electrical Coupling among (Myo)Fibroblasts and Myocyte

In addition to fibroblasts interacting with myocytes mechanically through the extracellular matrix and via chemical signaling, numerous studies have considered the effect of electrical coupling between these two populations. Both fibroblasts and myofibroblasts produce connexins and form gap junctions with neighboring cells within the myocardium. The current evidence supports homocellular, as well as heterocellular gap junctions forming within (myo)fibroblasts and myocytes [22,37,38,39,40,41,42]. As gap junctional communication facilitates the spread of de- and repolarization, coupling between and among (myo)fibroblasts and myocytes raises the intriguing question of how such coupling might influence the myocytes’ electrophysiology.

Unlike cardiac myocytes, cardiac (myo)fibroblasts are not excitable cells. They express a number of voltage- and ligand-gated, mechanosensitive, store-operated and transient receptor potential (Trp) channels mediating sodium, potassium, calcium and chloride currents, which modulate membrane potential and cell function (for recent comprehensive reviews, see Ross and Jahangir [43] and Feng et al. [44]). The effect of electrical coupling between excitable myocytes and non-excitable fibroblasts may be neutral or protective in a healthy heart, but with cardiac pathologies such as volume overload, ischemia, inflammation and elevated mechanical strain, may become relevant in developing a substrate for dysrhythmias.

Coupling between myocytes and (myo)fibroblasts has been predicted to act as current sinks [11,23,24,25,26,45], passively reducing myocyte excitability. This is believed to occur as depolarization is shared not only by the sarcoplasm of the myocyte, but also of electrically coupled (myo)fibroblasts. Electrical charge is thereby distributed over a larger volume than the individual myocyte and depolarization within the myocyte to the threshold for an action potential requires more charge to achieve. Such capacitive drain critically depends on the ratio of (myo)fibroblasts to myocyte, as well as the currents flowing in the (myo)fibroblasts [10,23,25]. The largest effect is predicted to occur in scar tissue following infarction, where surviving myocytes may be in small number relative to the surrounding myofibroblasts [10]. This passive electrical ‘drain’ may contribute to delayed atrioventricular conduction and the development of unidirectional blocks [42], both of which destabilize the electrical fabric of the myocardium and predispose to development of re-entry loops.

Heterocellular coupling between (myo)fibroblasts with myocytes has been demonstrated under culture conditions by dye coupling, loss of excitability in coupled myocytes when a large current was selectively activated in co-cultured myofibroblasts [10] and passive transfer of action-potential-like depolarization across strands of myofibroblasts separating myocytes [20,21]. Coupling of (myo)fibroblasts and myocytes was associated with reduced action potential duration, resting depolarization and induction of delayed after-depolarizations as well as heterogeneous current conduction [46,47]. Such electrical perturbations are predicted to both trigger and perpetuate atrial fibrillation. Similarly, co-culture experiments with myofibroblasts and myocytes isolated from pig ventricles demonstrated that myofibroblast coupling shortened action potential duration but was associated with hyperpolarized myocyte resting membrane potential [48]. In contrast to previous studies, fibroblasts co-cultured with myocytes did not affect myocyte electrophysiology in this model. These results suggested that heterocellular coupling may only promote dysrhythmias in pathological conditions where myofibroblasts are expressed [48]. Both the hyperpolarized and depolarized resting membrane potentials in co-cultured myocytes were predicted to produce dysrhythmias.

Demonstrating coupling in vivo has been more difficult. In an elegant study using selective optogenetic expression of voltage-sensitive fluorescent dye in fibroblasts, myofibroblasts or myocytes, Quinn et al. [19] provided direct evidence of electrical coupling between myocytes and non-myocytes at the border of healing cryo-induced injuries in mouse hearts. These authors also identified connexins where myocytes were in contact with non-myocytes. Using electron microscopy, tunneling nanotubes were identified, connecting cells at the border of the scar. Both the tunneling nanotubes and gap junctions were predicted to electrotonically couple myocytes to non-myocytes. Interestingly, no dysrhythmias were shown in representative optical pseudo-ECGs recorded from myocytes in the unwounded area or from non-myocytes in the wounded epicardium [19].

Given that the atria have higher concentrations of fibrous tissue relative to the myocyte layers [16], electrotonic coupling may have more impact in these chambers. Using electron microscopy to study fibroblasts within the sino-atrial node of rabbits, De Mazìere et al. [16] found homocellular coupling by gap junctions but little evidence of heterocellular coupling. The extent and consequence of heterocellular coupling within the heart remains an intriguing question.

In the fibrotic myocardium, electrical spread across both excitable myocytes and non-excitable (myo)fibroblasts may represent a secondary mechanism to ensure that the endo- and perimysium do not undermine the electrical syncytium. In the healthy myocardium, this is not likely to be an issue, as the extracellular matrix is small relative to the myocyte volume. However, in atrial fibrillation, reactive and reparative fibrosis may impair myocyte-to-myocyte coupling and so, secondary coupling via (myo)fibroblasts may be of particular importance.

Elegant studies by Miragoli, Gaudesius, Thomas and Rohr [20,21] have investigated the extent to which passive conduction of depolarization across (myo)fibroblasts may ensure action potential conduction across fibrous tissue. These scientists created an in vitro culture system in which cardiac myocytes were separated by inserts of cardiac fibroblasts with widths ranging from ~50 μm to 800 μm [20]. Using optical mapping of membrane potential with voltage-sensitive dyes, conduction of depolarization between bands of myocytes was investigated. This study confirmed that cultured cardiac fibroblasts do not generate action potentials, but that bands of up to 300 μm distance were successful in passively conducting a large enough wave of depolarization to evoke action potentials in the myocyte band beyond the fibroblasts [20]. However, the passive transmission of depolarization was slower in the fibroblast insert. Similarly, myofibroblasts were shown to be unexcitable, to couple with myocytes in culture and to passively conduct waves of depolarization stimulated from action potentials in coupled myocytes [21]. These data suggest that passive electrical coupling of (myo)fibroblasts within the fibrotic myocardium may preserve waves of action potentials between separated myocytes, but at the cost of heterogeneous conduction velocity. Heterogeneous action potential conduction in situ, from healthy myocardium through the peri-infarct region and across infarction scars in a rabbit model of apical infarction has also been demonstrated [49]. Heterogeneous action potential conduction velocity within the atrium is associated with perpetuation of atrial fibrillation [1].

These findings strongly suggest that under pathological conditions such as fibrosis and inflammation, electrotonic interactions between myocytes and (myo)fibroblasts may contribute to initiation and propagation of re-entry circuits, contributing to atrial fibrillation [11,34]. However, (myo)fibroblasts are not the only non-myocyte capable of interacting with and modulating myocyte excitability. Adipose tissue lying directly above and infiltrating into the myocardium may be another source of electrical disruption.

## 3. Epicardial Adipose Tissue, Fibrosis and Atrial Fibrillation

Obesity is a known risk factor for the development and progression of atrial fibrillation [50,51]. Epidemiology studies indicate that the extent and condition of epicardial adipose tissue are positively correlated with atrial fibrillation [14,51]. Surgical ablation of epicardial adipose tissue in the left atrium was successful in treating persistent atrial fibrillation [52]. Weight loss was associated with reduction in epicardial adipose tissue thickness [53] and when coupled with management of risk factors including hypertension and coronary artery disease, was associated with reversal of persistent to paroxysmal to no atrial fibrillation in 88% of participants [51]. These findings indicate that adipose tissue is also of interest when understanding and managing atrial fibrillation.

### 3.1. Epicardial Adipose Tissue

Epicardial adipose tissue is found between the visceral layer of the serous pericardium and the epicardial surface of the heart, in intimate contact with the myocardium [54]. It is distinct from paracardial adipose tissue, which is located within the visceral and parietal layers of the serous pericardium rather than against the myocardium [55]. Together, epicardial and paracardial adipose tissues comprise the pericardial fat deposit [6,53]. Of note, the nomenclature concerning cardiac adipose tissue is not consistent within the literature and epicardial adipose tissue has also been referred to as intra-thoracic, mediastinal, paracardial and pericardial [55]. Within the heart, epicardial adipose tissue is sometimes named for different locations, as with peri-atrial, peri-ventricular and perivascular/peri-coronary epicardial adipose tissues [56,57]. As each fat deposit may have distinct transcriptomic signatures [57], it is best to check the anatomical definitions within research papers to be sure of which fat bodies are being discussed.

Epicardial adipose tissue varies across species as well. In small mammals such as rats (Figure 2A,B) and mice, epicardial adipose tissue is minimal [58]. In humans (Figure 2C) and other large mammals, epicardial adipose tissue surrounds the atria and is associated with the coronary vasculature on the epicardial surface of the ventricles, comprising 15–20% of total weight of human hearts [53] and covering 56–100% of the cardiac circumference [59,60,61]. In addition to perivascular fat deposits which appear to protect the coronary vasculature [61], epicardial adipose tissue also covers large stretches of the ventricular epicardium in healthy individuals [58]. The extent and thickness of the epicardial adipose tissue increases as body–mass index increases through over-weight and obese in larger mammals such as humans [60,62]. Both obesity and increased epicardial adipose tissue volume/thickness are risk factors for cardiovascular disease including atrial fibrillation [51,63,64].

Epicardial adipose tissue is home to adipocytes, preadipocytes, fibroblasts and in healthy human hearts, a small population of macrophages [53,65]. It shares a common embryological origin as cardiac fibroblasts and myocytes and a common blood supply [55,61]. Numerous studies have demonstrated that epicardial adipocytes interact with myocardial fibroblasts, leukocytes and myocytes where epicardial adipose tissue is in direct contact with the myocardium [61,66].

In addition to lying along the epicardial surface of the myocardium, epicardial adipose tissue is also found within the myocardium as fatty infiltrates. Even in cardiac samples from healthy human and other large mammals, fatty infiltrates of the myocardium appear common [58]. In atrial appendages obtained from patients with atrial fibrillation, the extent of epicardial fat infiltrates was positively correlated with the severity of myocardial fibrosis, with numerous patients showing transmural infiltration of epicardial adipose tissue [13,14]. The roles of epicardial adipose tissue and in particular, the fatty infiltrates in health and their changes in disease such as atrial fibrillation and obesity are not yet fully understood. However, the presence of adipose tissue within the myocardium is of potential significance to the electrical profile and health of the myocardium.

Epicardial adipose tissue is thought to play a number of important roles in the healthy heart. It provides a quick source of energy for the myocardium, as well as reducing the risk of lipotoxicity [61]. Epicardial adipose tissue is also believed to offer mechanical protection. It appears to cushion the heart from a strike to the sternum, which may trigger potentially fatal cardiac dysrhythmias. It may also protect the coronary vasculature against torsion during myocardial contraction and pulse wave propagation. Such protection may also extend to nerves; epicardial adipose tissue contains autonomic nerves and ganglionated plexi [67]. Epicardial adipose tissue assists with immunological support and provides both insulation and heat production to help thermoregulate the heart [61].

**Figure 2 cells-10-02501-f002:**
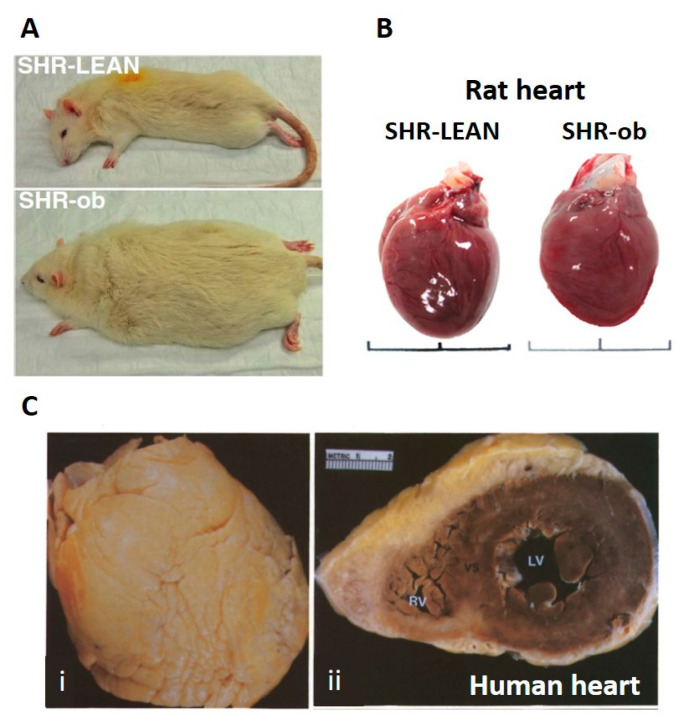
Comparison of rat and human hearts, showing deposition of epicardial adipose tissue in human hearts, but not in rat hearts. Panel **A**: lean vs. obese (ob) spontaneously hypertensive rats (SHR), adapted from Linz et al. [68]. Panel **B**: hearts from lean vs. obese SHR. Note the absence of epicardial adipose tissue on each heart, regardless of animal weight. Adapted from Shiou et al. [69]. Panel **C**: heart of an 83-year-old woman (body–mass index 25.3; overweight) illustrating extensive epicardial adipose tissue covering the anterior surface (i). In cross-section (ii), the thickness of the adipose layer over the anterior aspect of heart is evident, while a thinner layer covers the posterior aspect of the heart. Adapted from Shirani et al. [70].

### 3.2. Fat Phenotype and Coupling via Gap Junctions

The thermogenic nature of epicardial adipose tissue is an intriguing aspect of its function. Adipose tissue is generally classified as brown or white, depending primarily on whether the adipocytes participate in thermogenesis. Brown adipose tissue is commonly found in infants and lost as the individual ages [71].

Another phenotype has been described, ‘beige’ or ‘brite’ (‘brown-in-white’) adipose tissue, in which white adipocytes express ‘brown’ genes and characteristics, including uncoupled respiration and increased thermogenic potential [56,57,72,73,74,75]. Beige/brite adipose tissue is observed when brown-like adipocytes are found within white adipose tissue, thought to be in response to cold challenge [72,75,76] or activation of sympathetic adrenergic receptors on adipocytes [77]. This transition is reversible, with beige/brite adipocytes resuming a white-like phenotype when the challenge is removed [77]. Intriguingly, epicardial adipocytes from healthy human adults are considered to express brown [55] or ‘beige/brite’ [55,74] characteristics, in that the epicardial adipocytes express the mitochondrial uncoupling protein UCP1 and display associated uncoupled respiration [78], hallmark features of brown adipocytes that allow for non -shivering thermogenesis [74].

Aging and disease states appear associated with a phenotypic change from brown to white adipose tissue, which is linked to inflammation and lipid dysregulation [71,74]. For example, epicardial adipose tissue may undergo a brown/beige-to-white transition during obesity, which may contribute to the pathological changes and development of cardiovascular diseases such as atrial fibrillation [55]. “Re-browning” of epicardial adipose tissue by upregulation of brown-adipocyte-specific genes was associated with reduced inflammation and has been proposed as a therapeutic goal in managing cardiovascular disease [79].

An intriguing question is whether the intimately associated epicardial adipocytes may form homocellular gap junctions within adipose tissue (both surface and the infiltrates) and heterocellular gap junctions with myocytes and/or fibroblasts within the adjacent myocardium. If so, then the epicardial adipose tissue may directly contribute to the creation of a heterogenous electrical environment in atria at risk of atrial fibrillation. Notably, gap junctions have been demonstrated in epicardial fat in hamsters [80,81]. Moreover, in brown and beige/brite fat, adipocytes are functionally coupled by gap junctions; indeed, upregulated expression of connexin 43 is important in white adipose tissue conversion to beige/brite adipose tissue [76,82,83]. In contrast, gap junctional coupling in white adipose tissue is reduced compared to brown or beige/brite adipose tissue [82]. Burke et al. [82] considered white adipose tissue from the epididymis and mesentery and brown adipose tissue from the interscapular fat pad in adult mice; whether these findings apply to human adipose tissue and/or epicardial adipose tissue are important questions in the context of atrial fibrillation.

### 3.3. Epicardial Adipose Tissue Secretome

In addition to being in direct contact with (myo)fibroblasts and myocytes within the myocardium, epicardial adipose tissue also participates in chemical communication. Adipokines within the epicardial adipose tissue secretome modulate leukocytes and fibroblasts within the adipose tissue as well as leukocytes, fibroblasts and myocytes in the associated myocardium. Numerous adipokines such as omentin, apelin and adiponectin are anti-fibrotic and anti-inflammatory, while others such as resistin, adipose fatty-acid-binding protein and visfatin are pro-inflammatory. Leptin may be protective or associated with pathological changes (for recent reviews see Antonopoulos and Antoniades [61], Oikonomou and Antoniades [66] and Smekal and Vaclavik [84]). The balance between pro- and anti-inflammatory adipokines is labile, shifting away from a dominance of protective paracrine factors to a pathological dominance of pro-inflammatory elements. The pathological condition is associated with inflammation, fibrosis and generation of free oxygen radicals [6].

It is worth noting, however, that in humans with heart disease and a predisposition for atrial fibrillation compared to individuals with sinus rhythm in case-controlled studies, atrial epicardial adipose tissue upregulates the expression of some pro-inflammatory and pro-fibrotic adipokines while down regulating the expression of anti-fibrotic cytokines (Table 1 and references therein). These observations may contribute to paracrine signaling that shifts the myocyte to fibroblast ratios in the atria over time. This may then provide a substrate for the development of dysrhythmias. Detailed genetic, epigenetic and pathophysiological factors that cause these changes in transcription and secretory function in epicardial adipose tissue are yet to be fully elucidated.

When considering chemical communication between epicardial adipose tissue and the myocardium, the neural network within the adipose tissue is also of note. Autonomic nerves and ganglionic plexi carried within the peri-atrial fat [70] release adrenergic neurotransmitters, acetylcholine and vasoactive intestinal protein (VIP) [85]. Atrial fibrillation is associated with chronically elevated sympathetic activity, which is observed in obesity [86]. It is possible that neurotransmitters produced by nerves within epicardial adipose tissue may also modulate myocyte excitability and propensity to produce triggered activity, contributing to atrial fibrillation. Lin et al. [86] investigated this possibility in a rabbit model. Application of isoproterenol to mimic such elevated sympathetic stimulation in rabbit left atrial myocytes was associated with significantly more delayed afterdepolarizations when myocytes were co-cultured with epicardial adipose tissue, compared to monocultures of myocytes [86].

**Table 1 cells-10-02501-t001:** Examples of changes in adipokines produced by epicardial adipose tissue in patients with atrial fibrillation, compared to those with sinus rhythm.

Study Reference	Quantified Biomarker in EAT, Method Used and Known Effects on Inflammation or Fibrosis	Effects on Atrial Fibrillation Phenotype and Nature of Controlled CASE Study
Chen et al. [87]	Omentin-1 (protein expression). Inhibits transforming growth factor (TGF)-β1-induced fibrosis.	Down regulated in patients with heart valvular disease and atrial fibrillation, (*n* = 4–20)
	TGF-β1 (protein expression). Controls cell differentiation promoting fibrosis.	Up regulated in patients with heart valvular disease and atrial fibrillation (*n* = 4–20).
Wang et al. [88]	cTGF (mRNA and protein expression). A central mediator of tissue remodeling and fibrosis.	Up regulated in coronary bypass surgery patients with atrial fibrillation, (*n* = 16).
	Leptin (mRNA and protein expression). Stimulates inflammation, matrix metalloproteinase (MMP) activity and interstitial fibrosis.	Up regulated in coronary bypass surgery patients with atrial fibrillation, (*n* = 16).
	Vaspin (mRNA and protein expression). Improves insulin sensitivity and anti-fibrotic.	Up regulated in coronary bypass surgery patients with atrial fibrillation, (*n* = 16).
Wang et al. [89]	YKL-40/*CHI3L1*, (mRNA and protein expression). A novel biomarker for inflammation, tissue remodeling and fibrosis.	Up regulated in coronary bypass surgery patients with atrial fibrillation, (*n* = 28).
Kira et al. [90]	Collagen. Characteristic biomarker for collagen deposition & fibrosis.	Up regulated in a retrospective evaluation of autopsies for patients with atrial fibrillation, (*n* = 3).

Cholinergic nerves within epicardial adipose tissue have been studied in the context of post-operative atrial fibrillation, which is a common complication of cardiac surgery [91]. Injection of botulinum toxin into epicardial adipose tissue, blocking release of acetylcholine from the parasympathetic nerves, reduced post-operative fibrillation within 30 days following surgery. The protection was sustained over the year following surgery. The patients in this study had a history of paroxysmal atrial fibrillation and underwent coronary artery bypass surgery. The protective effect may lie in modulation of the epicardial adipose tissue secretome. Muscarinic cholinergic receptors were detected in adipocytes within epicardial adipose tissue samples from 85 patients during open heart surgery; stimulation of the adipose tissue with acetylcholine modified the secretome the adipose tissue produced [92].

### 3.4. Pathological Adipose Tissue, Inflammation and Fibrosis

In the pathological condition, the loss of anti-inflammatory and anti-fibrotic adipokines is associated with marked fibrosis within the epicardial adipose tissue as well as in associated myocardium [93]. In both right- and left-atrial samples from patients with or without atrial fibrillation (Abe et al. [13]: left atrial appendages from 59 patients with paroxysmal (46%) and persistent (54%) atrial fibrillation; Haemers et al. [94]: right atrial samples from 92 patients in whom 65% lacked atrial fibrillation, 20% had paroxysmal and 15% had persistent atrial fibrillation; Nalliah et al. [14]: right atrial appendages from 19 patients in whom no history of atrial fibrillation was present), the extent of fibrosis in the epicardial adipose tissue (both surface, Figure 3A, and infiltrates, Figure 3A–C) was positively correlated with fibrosis in the associated myocardium [13,14,94]. Furthermore, the extent of the fibrofatty infiltrates within the myocardium from human right atrial appendages was correlated with the type of atrial fibrillation (absent to paroxysmal to persistent), suggesting that the increasing level of fibrosis was associated with progression of atrial fibrillation [94]. Such patients’ atria show clear endomysial and perimysial fibrosis around the cardiac myocytes (Figure 1). As with reactive and reparative fibrosis associated with activation of fibroblasts to myofibroblasts, endomysial and perimysial fibrosis is predicted to promote ischemia and dysrhythmias. Furthermore, hypoxia within expanding subcutaneous and visceral adipose tissues in obesity triggers fibrosis and maladaptive changes in the adipocyte phenotype [95]. Whether this is true in epicardial adipose tissue is of interest, given that ischemic heart disease [96] and heart failure [97] are associated with increased risk of atrial fibrillation.

Development of fibrosis and atrial fibrillation was associated with a marked shift from fatty infiltrates to fibrofatty infiltrates within the myocardium and increased thickness in the surface adipose tissue layer [6,94]. Inflammation and fibrosis in adipose tissue are closely associated. Within adipose tissue, activated macrophages are key in promoting fibrosis. These cells secrete transforming growth factor (TGF)-β, which stimulates myofibroblast-like differentiation of preadipocytes. Increased fibrosis is associated with increased secretion of pro-inflammatory adipokines, which further promote adipose tissue fibrosis [98]. These adipokines appear to affect the associated myocardium as well: fibrosis in human left atrial appendage myocardium was more pronounced where the myocardium was in intimate contact with adipose tissue [13]. TGF-β secreted by macrophages within epicardial adipose tissue may have paracrine effects on fibroblasts within the atrial myocardium, promoting differentiation to myofibroblasts and fibrosis.

Another source of both adipocytes and fibroblasts within the myocardium are the thin layer of epicardial-derived progenitor cells which, upon acute injury, may become activated, move into the myocardium and differentiate into either fibroblasts or adipocytes [99]. Culture experiments of human atrial adult epicardial progenitor-derived cells with the atrial myocardial secretome was associated with differentiation of cells into both phenotypes (one third fibroblast; two-thirds adipocytes). However, incubation with fibrogenic or adipogenic media could shift the differentiation to fibroblasts or adipocytes, respectively. These data suggest that a shift in the secretome from anti-fibrotic to pro-fibrotic during disease development and progression may play a role in the development of fibrofatty infiltrates and myocardial fibrosis [99].

In sheep models of obesity and atrial fibrillation, obesity was associated with fibrosis in both epicardial adipose tissue and the left atrial myocardium as well as electrical disturbances [50,94,100]. Fibrofatty infiltrates and fibrosis in surface epicardial adipose tissue were associated with slower action potential conduction and greater conduction heterogeneity within the atrial myocardium [50,100]. Obesity was also associated with more frequent and longer bouts of atrial fibrillation, with greater cumulative duration [100].

### 3.5. Mechanistic Role of Fibrofatty Infiltrates in the Perpetuation of Atrial Fibrillation

It seems clear that fibrofatty infiltrates within the atrium are associated with increased risk of atrial fibrillation and with progression of fibrillation. Do pro-inflammatory and pro-fibrotic chemical communication within adipose tissue produce inflammation, fibrosis and/or electrical disturbances in the associated myocardium? Such chemical modulation is one of two basic models of dysrhythmogenic interactions between fat and muscle. The alternative model is direct electrotonic coupling of adipocytes and myocytes [14]. Both direct coupling through gap junctions and indirect modulation via adipokines have been proposed to underlie the electrical changes observed in atrial fibrillation.

Atrial fibrillation is associated with lateralization of myocyte connexin-40, resting depolarization of myocytes, heterogeneous action potential conduction velocity with areas of slower conduction and increased propensity to triggered activity and re-entry [14]. These electrical abnormalities provide an electrical substrate in which ectopic pacemakers may form, triggering re-entry loops and fibrillation. In left-atrial appendages from patients with atrial fibrillation (paroxysmal or persistent), the extent of fibrosis, lateralization of myocyte connexin-40 and conduction abnormalities were associated with the extent of epicardial fibrofatty infiltrates [13]. These data suggest that the presence of epicardial adipose infiltrates within the myocardium evoked myocardial remodeling which lead to atrial fibrillation [13,14]. This suggestion is further supported in a study of patients with atrial fibrillation undergoing ablation, who were stratified according to body–mass index [101]. In obese participants (body–mass index ≥ 27; 16 patients in whom 62.5% suffered paroxysmal atrial fibrillation and 37.5% suffered persistent atrial fibrillation), atrial myocardial conduction velocity was slowed and fractionated across the left atrium with regional reduction in voltage and increased low-voltage area, compared with non-obese participants (body–mass index < 27; 10 patients with 80% paroxysmal and 20% persistent atrial fibrillation) [101].

Epicardial adipose tissue is also modulated by the myocardium. In sheep models of obesity, fibrillation was associated with further remodeling of epicardial adipose tissue, suggesting that the fibrillating myocardium also modulates the associated adipose tissue [94]. Such reciprocal, pathological remodeling in a positive feedback loop is consistent with the observation that the presence of atrial fibrillation promotes further fibrillation (“atrial fibrillation begets atrial fibrillation”) [102].

In vitro patch-clamp studies of rabbit left but not right atrial myocytes co-cultured with epicardial adipose tissue indicated that the presence of adipose tissue was associated with myocyte action potential prolongation; resting depolarization; higher incidence of triggered activity (delayed afterdepolarizations); increased late sodium current and L-type calcium current; and reduced transient outward, delayed rectifier and inwardly rectifying potassium currents. Treatment with conditioned media from epicardial adipose tissue produced similar but less pronounced changes to the myocyte electrophysiology. Intriguingly, these authors noted that myocytes were rarely observed in direct contact with adipocytes in co-culture. This observation suggests that the enhanced effect of co-culture was not due to direct electrotonic interactions [90]. Heart failure enhanced the effect of epicardial adipose tissue on the myocytes: microelectrode studies of rabbit atria indicated that the effect of epicardial adipose tissue on action potential duration and spontaneous dysrhythmias in left atrial myocytes was more potent when heart failure was induced by 4 weeks of high frequency right ventricular pacing [89]. Similarly, in vitro studies of human induced pluripotent stem cell-derived cardiac myocytes cultured with sheep adipose tissue indicated that myocyte beat frequency slowed while field potential duration increased [14]. These studies suggest that chemical modulation of myocyte electrophysiology is sufficient to produce a substrate for atrial fibrillation in these in vitro models. In vivo studies will be required to explore the relative importance of chemical and electrotonic modulation of myocyte electrophysiology by the associated epicardial adipose tissue in patients with atrial fibrillation.

Of relevance to this discussion are studies of adipocyte-derived stromal multipotent cell therapy in hearts damaged by myocardial infarction. Such stem cell therapy is promising in restoring myocardial function but holds a risk of evoking dysrhythmias through chemical or electrotonic interactions with myocytes. Using monolayers of cultured rat neonate ventricular myocytes, ten Sande et al. [103] tested the effect of either co-culture with adipocyte-derived stromal cells or conditioned media from separate cultures of adipocyte-derived stromal cells. The adipocytes used as the source of the stromal cells were obtained by subcutaneous fat aspirates from rat, pig and human donors. Significant slowing of conduction velocity was seen in myocytes in all co-cultures. Only conditioned media from stem cells derived from pig subcutaneous adipose tissue slowed conduction velocity in myocytes in mono-culture; in adipocyte-derived stromal cells from human and rat fat, conditioned media had no effect. With stromal cells from these species, co-culture of fat and muscle was required to modulate myocyte electrophysiology. These data suggest that both chemical and direct electrotonic interactions may underlie the abnormal electrical substrate associated with fibrillation, with species differences [103].

## 4. Conclusions, Challenges and Future Directions

Atrial fibrillation is associated with fibrosis, changes in myocyte action potential duration, heterogeneity of conduction velocity, generation of triggered activity and re-entry loops. Pathological changes in epicardial adipose tissue and differentiation of fibroblasts to myofibroblasts are associated with fibrosis and are likely to play a role in the pathogenesis of atrial fibrillation associated with fibrosis.

Numerous studies suggest that myocytes may interact chemically and electronically with non-myocytes. Key non-myocyte populations in the atria are (myo)fibroblasts and epicardial adipose tissue (containing adipocytes, pre-adipocytes which may differentiate into myofibroblast-like cells, a neural network and leukocytes)(Figure 4A). In silico, in vitro and in vivo studies of human tissue and in non-human animal models suggest that both adipocytes and (myo)fibroblasts modulate myocyte electrophysiology either indirectly (by chemical communication and/or by changing the architecture of the myocardium) or directly (by electrotonic modulation) (Figure 4B).

### 4.1. Challenges in Studying the Pathophysiology of Atrial Fibrillation

Numerous challenges arise when investigating the mechanistic unpinning of atrial fibrillation. One challenge is that atrial fibrillation may take many forms, which may reflect distinct underlying pathophysiologies. Studies of human patients suffering from atrial fibrillation generally include those experiencing paroxysmal or persistent atrial fibrillation. These participants are commonly grouped without distinction being made of which type of fibrillation the patients were experiencing. Furthermore, variability among human participants reduces the ability of studies to delineate clear mechanistic cause and effect, compared to non-human animal models.

Another challenge lies in the choice of the correct non-human animal model for mechanistic studies. Small animal models may be useful when studying cardiac physiology. In particular, murine models are relatively low cost, readily available and have the potential for genetic manipulation through the production of transgenic mice [106,107]. Rat and mouse models in general are the most versatile and have well established relevant techniques such as electrophysiology. They have been employed in studies considering the pathways and mechanisms involved in atrial fibrillation, but are considered limited in their use [106,108]. The disadvantages of using these rodent models for human atrial fibrillation are substantial, including their heart rate which varies considerably compared to humans, their heart size, the need to induce atrial fibrillation by electrical stimulation and the differences in atrial and ventricular action potentials in rats and mice, when compared to humans [107,109]. These animal models can also lack the comorbidities which are often associated with atrial fibrillation and so do not accurately represent the pathophysiology of atrial fibrillation in humans.

Guinea pigs and rabbits have also been used and are considered to be a better representation of human atrial fibrillation. However, larger mammals such as goats, sheep and pigs in particular have hearts which more closely resemble the human heart anatomical structure, including more extensive epicardial adipose tissue (Figure 2). Hearts of these larger mammals, goats in particular, may develop prolonged and sustained atrial fibrillation, making them useful for studies investigating the structural and electrical remodeling, [107,110] and the electrophysiological changes [111]. Horses, interestingly, develop atrial fibrillation naturally and show atrial remodeling that is similar to humans [112], but have different electrophysiology from humans.

While atrial fibrillation in these larger species’ hearts have closer resemblances to human atrial fibrillation, there are limitations for their use such as cost, housing and ethical constraints [107]. All of these models are valuable for a targeted approach investigating specific aspects of atrial fibrillation such as acute atrial insults, atrial structural remodeling, rate-related modelling, autonomic vagal nerve stimulation in the larger animal models and genetic manipulation to induce specific characteristics of atrial fibrillation and myocardial infarction [113]. Overall, there is no ideal animal model that truly reflects human dysrhythmias and so translation of the results to the clinical setting can be challenging, but could pave the way for clinical trials for novel therapeutics and devices.

### 4.2. Future Directions

There are many questions yet to be answered. The exact phenotype of epicardial adipose tissue varies with location in the heart, age and disease. It appears to have a brown/beige/brite phenotype in health and switches to white adipose tissue with age and disease. The phenotype is relevant as brown/beige/brite adipose tissue express more gap junctions, when compared to white adipose tissue. It is not clear whether adipocytes are capable of forming functional gap junctions with adjacent (myo)fibroblasts and/or myocytes. If so, then electrotonic modulation of myocyte electrophysiology is likely, similar to that observed with (myo)fibroblasts and myocytes. If heterocellular coupling does exist with adipocytes (as proposed in Figure 4B), the relative volume of adipocytes to myocytes is relevant. As non-myocytes act as a passive current sink, the larger the capacitance of non-myocytes coupled with a myocyte, the larger the passive sink and the more difficult it will be for the myocyte to depolarize to the threshold for action potential production. Both healthy and pathological adipocytes (Figure 1 and Figure 2) are very large in comparison to myocytes, which are in turn very large in comparison to (myo)fibroblasts. This suggests that coupled adipose tissue may represent a significant current sink in the atrium. However, the presence of fat within the adipocyte may reduce the conductive volume significantly, mitigating the effect of passive electrotonic coupling.

The possibility and implications of heterocellular coupling among adipocytes, myocytes and/or (myo)fibroblasts could be studied in co-culture experiments similar to those of Gaudesius et al. [20] and Miragoli et al. [21]. However, adipose tissue poses a distinct challenge when attempting to co-culture it with myocytes: it tends to float and may not intimately interact with a monolayer of myocytes. Consequently, more sophisticated 2D and 3D culture systems [39,114,115] may be important if the question of whether adipocytes might electronically couple myocytes is investigated.

It is also clear that in vitro systems cannot easily replicate the in vivo complexity of adipose tissue (containing adipocytes, nervous tissue, leukocytes and proto-adipocytes), fibrous tissue with attendant (myo)fibroblasts and myocytes. Optogenetic studies of atria as well as microelectrode studies of fresh intact atrial tissue may provide a way forward, in both human biopsies and non-human animal models of atrial fibrillation with comorbidities such as obesity.

Another key direction for future studies is to determine what controls the switch in epicardial adipose tissue from cardioprotective (anti-inflammatory and anti-fibrotic) to pathological (pro-inflammatory, pro-fibrotic and pro-free oxygen radicals). This phenotypic switch is likely to vary with comorbidity (obesity, diabetes, ischemic heart disease, valve disease). For example, it may occur more rapidly in patients with obesity and diabetes, compared to diabetes or obesity alone. The complex interplay between electrical, mechanical/architectural and chemical interactions between adipose tissue, myocytes and (myo)fibroblasts is likely to vary with such comorbidities, as well as showing regional (right vs. left atrial vs. ventricular vs. perivascular) differences.

Two avenues of research which will help delineate the complex interplay and changes associated with fibrosis, non-myocytes and myocytes in development and perpetuation of atrial fibrillation are genetic and epigenetic studies. The role that genetics play in atrial fibrillation is gaining momentum and candidate genes are being identified in which mutations and polymorphisms such as single nucleotide polymorphisms (SNPs), are associated with atrial fibrillation [116,117].

Most studies identifying candidate genes have been in patients experiencing lone atrial fibrillation with no underlying disease [118,119]. These candidate genes have been identified mostly through genome wide association studies (GWAS) and familial studies demonstrating that atrial fibrillation can be inherited [116,120,121]. Many of these genes identified have been linked to genes that code for ion channels, or are involved in fibrosis nuclear structure, cardiogenesis, extracellular matrix remodeling and cell–cell coupling [122]. The genetics of atrial fibrillation can provide new insights into disease processes and novel therapeutic targets, however, recent research is also exploring the role epigenetics plays in the pathophysiology of atrial fibrillation.

Epigenetics is another mechanism by which gene expression is controlled and subsequently can play a role in diseases [123]. Of the different types of epigenetic regulation, histone modifications (acetylation, methylation, phosphorylation and ubiquitination), DNA methylation and microRNAs are the most studied [124,125]. Changes in the epigenetic landscape within a cell are heritable and these changes can cause dysregulation in gene expression, leading to disease. Such changes are now thought to play a role in the onset and progression of atrial fibrillation through the mis-regulation of atrial fibrillation-associated genes and subsequent structural and electrical remodeling [126,127]. More recently the transcriptome and proteome are also being analyzed, providing more information on the potential pathways that are involved in pathogenesis of atrial fibrillation [128]. These pathways can also be used to target genes that might be involved in pathogenesis of atrial fibrillation and can be used as therapeutic targets and potential biomarkers. Biomarkers for atrial fibrillation are already being identified [129,130] and could be useful in clinical risk assessment and outcomes for individuals with atrial fibrillation.

Understanding these complex interactions may help understand the initiation and progression of atrial fibrillation. Adipose tissue and fibroblasts may represent important therapeutic targets in managing atrial fibrillation and preventing disease progression.

## Figures and Tables

**Figure 1 cells-10-02501-f001:**
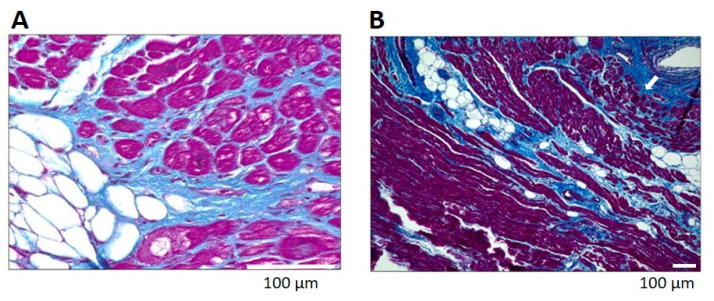
Fibrosis in biopsies of left atria from patients with atrial fibrillation. Note pronounced perimysial and endomysial fibrosis (blue) in the subepicardial (Panel **A**) and perivascular (Panel **B**, arrow) myocardium. Masson’s Trichrome staining, adapted from Abe et al. [13].

**Figure 3 cells-10-02501-f003:**
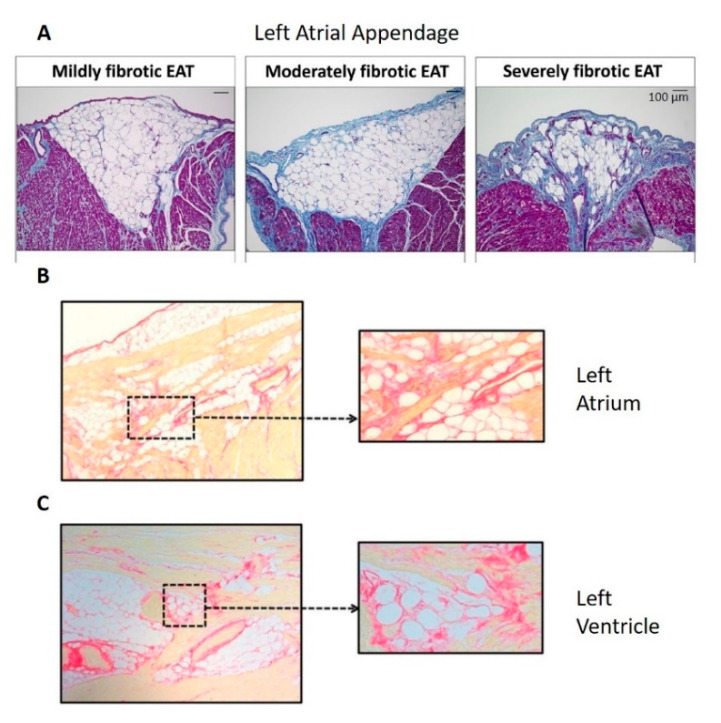
The extent of fibrosis within the epicardial adipose tissue (EAT) was positively correlated with the extent of myocardial fibrosis (Panel **A**), within human left atrial appendage (Panel **A**), left atrium (Panel **B**) and left ventricle (Panel **C**). Panel **A**: Masson’s Trichrome staining in which collagen is blue, adapted from Abe et al. [13]; panels **B** and **C**: Picrosirius Red staining in which collagen is red, adapted from Venteclef et al. [93]. No scale was provided for panels **B** and **C** in the original paper.

**Figure 4 cells-10-02501-f004:**
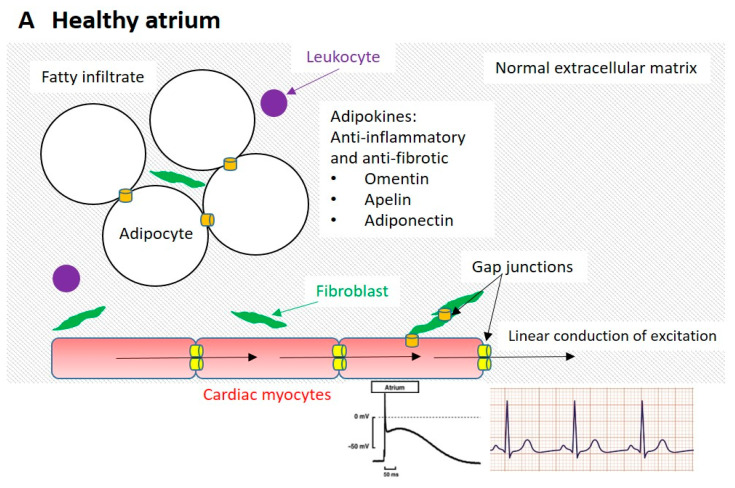
Proposed physical, chemical and electrotonic interactions between atrial myocytes and non-myocytes in atrial fibrosis and fibrillation. Panel **A**: healthy condition, in which the extracellular matrix is minimal in both epicardial adipose tissue and the myocardium. Fibroblasts are present, with few leukocytes. Adipocytes secrete an anti-inflammatory profile of adipokines. Cardiac myocyte gap junctions are concentrated at intercalated discs, allowing linear flow of excitation. The myocyte action potential and ECG are normal. Action potential adapted from Yacoub et al. [104]; normal ECG from www.aclsmedicaltraining.com, accessed on 21 August 2021. Panel **B**: fibrotic atrium suffering from atrial fibrillation, in which marked fibrosis is observed in the epicardial adipose tissue and myocardium. The epicardial fatty infiltrate is more extensive and has progressed to a fibrofatty infiltrate with abnormal adipocytes, myofibroblast-like cells and more numerous leukocytes. The adipokine profile is pro-inflammatory and pro-fibrotic and associated with increased production of reactive oxygen species. Fibroblasts have differentiated into myofibroblasts and myocyte to non-myocyte electrotonic coupling is more extensive. In the myocytes, lateralization of gap junctions produces zig-zag conduction. Action potential duration is prolonged, producing early after-depolarisations (EADs) which may trigger re-entry and fibrillation. Action potential adapted from Zhabyeyev and Oudit [105]; ECG showing atrial fibrillation from www.medicine-on-line.com, accessed on 21 August 2021.

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
