# Peer review of "Are Interactions between Epicardial Adipose Tissue, Cardiac Fibroblasts and Cardiac Myocytes Instrumental in Atrial Fibrosis and Atrial Fibrillation?"

_cells, 2021, doi:10.3390/cells10092501_

Round 1

Reviewer 1 Report

Most often atrial fibrillation is associated with complex alterations of the atrial myocardium with fibrosis, dystrophic myocardium and adipose tissue infiltration as mean features. Among the various mechanisms that can link such a tissue remodeling and the arrhythmia, abnormal coupling between cardiomyocytes and other cell types, fibroblast and adipocytes, has emerged as an important one. The article of Krishnan et al. reviews evidence for a coupling between fibroblast, adipocyte and atrial myocytes in the setting of atrial fibrillation. The first chapter is dedicated to fibrosis and fibroblast/myocyte coupling for which solid evidences exist for functional coupling between fibroblast and cardiomyocyte. The second part of the article addresses the question of a coupling between adipocytes and atrial myocytes, more speculative.   

              The first consequence of adipose tissue infiltration of the myocardium is the “uncoupling“ effect between myocyte dur to the fact that fat tissue works as an electrical isolation, however, this point is not really addressed.

              Well established coupling mechanism between fat and myocytes are the adipokines secreted by the epicardial adipose tissue. For instance, Actine A has been shown to induce the fibrosis of the neighboring myocardial tissue by stimulating myofibroblast notably in patient with heart failure or diabetes (ref#93). However, this study is not referred, see table 1.

              Recently it has been shown that the epicardial layer that contains resident progenitors  can give rise to adipocyte or fibroblast depending on distinct stimuli, thereby contributing to the fibro fatty infiltration of the atrial subepicardium and the electrical disconnection with the rest of the myocardium (PMID: 32175811). This study is not discussed.  

Author Response

Responses to Reviewer #1

We would like to thank the reviewer for their thoughtful review.

  1. “Moderate English changes required”

Response: Thank you for this feedback. The manuscript has been very carefully proof-read and edited to improve the language where changes appeared appropriate. The largest change is deletion of all material from lines 496 to 535, where it was discovered that the material in Section 3.4 had been repeated as Section 3.5. The number of the subsequent section on line 536 was amended from 3.6 to 3.5 in light of this deletion. Additional changes include amending ‘secretosome’ to the more correct term ‘secretome’ (lines 369, 372, 434 and 436) and ‘transcriptosome’ to ‘transcriptomic signatures’, which is the term used in the relevant reference (line 260). Finally, minor grammatical or wording changes were made on lines 146, 157, 176,  245, 336, Table 1 right-hand heading, lines 449, 541, and 694 to improve the quality and consistency of the prose.

  1. The article of Krishnan et al. reviews evidence for a coupling between fibroblast, adipocyte and atrial myocytes in the setting of atrial fibrillation. The first chapter is dedicated to fibrosis and fibroblast/myocyte coupling for which solid evidences exist for functional coupling between fibroblast and cardiomyocyte. The second part of the article addresses the question of a coupling between adipocytes and atrial myocytes, more speculative.

Response: Thank you for this comment. Yes, we agree that many of the assertions on the coupling between myocytes, adipocytes and (myo)fibroblasts and, the extent of its functionality in vivo are equivocal. To this effect, we have cited reviews and primary literature that provide a balanced approach to these arguments. For example, the review by Kohl P and Gourdie, doi: 10.1016/j.yjmcc.2013.12.024, does outline the extensive knowledge gaps in this area. Similarly, in the section on conclusions and future directions, we highlighted the need for more elucidative studies on the issue of electric coupling between myocytes and non-myocytes and the putative link to development of atrial fibrillation.

  1. The first consequence of adipose tissue infiltration of the myocardium is the “uncoupling“ effect between myocyte dur to the fact that fat tissue works as an electrical isolation, however, this point is not really addressed.

 Response: Thank you for this comment. This is a pertinent point, but we purposely did not delve too much into this aspect because we think it is a bit peripheral to the central theme of this review. Our primary objective was to explore potential links and mechanisms between epicardial adipose tissue deposition, fibrosis and atrial fibrillation. Nonetheless, in the revised version of our manuscript (Introduction, page 2, lines 71-73), we now have included an argument to support the notion that imbalanced adipocyte infiltration in myocytes may create a general electrical conduction heterogeneity due to an anatomical blockade; this causes slowing and anisotropy in the conduction of electrical impulses. (doi: 10.1016/j.hrthm.2018.06.025; doi: 10.3389/fphys.2018.00002; doi: 10.1111/pace.13825). 

  1. Well established coupling mechanism between fat and myocytes are the adipokines secreted by the epicardial adipose tissue. For instance, Actine A has been shown to induce the fibrosis of the neighboring myocardial tissue by stimulating myofibroblast notably in patient with heart failure or diabetes (ref#93). However, this study is not referred, see table 1.

Response: Table 1 was not meant to be an exhaustive outline of all pro-fibrotic adipokines but rather highlight changes in pro-fibrotic adipokines in epicardial adipose tissue that have also been directly linked with phenotypic manifestation of atrial fibrillations compared to patients with sinus rhythm in case-controlled studies. We do acknowledge that a plethora of pro-inflammatory and pro-fibrotic adipokines are associated with maladaptive changes including the development of atrial fibrillations and this was the essence of citing reference #93 and others in the original submission. Table 1 was meant to capture the potential paracrine effects of epicardial adipocytes and thus purposely excluded adipokines such as activin, adiponectin and IL-6 that were quantified in plasma or in ex vivo settings, much as they have a strong correlation with myocardial fibrosis. However, throughout the manuscript, we have cited relevant reviews and primary literature that highlights the role of adipokines in inflammation and fibrosis more generally.   

  1. Recently it has been shown that the epicardial layer that contains resident progenitors can give rise to adipocyte or fibroblast depending on distinct stimuli, thereby contributing to the fibro fatty infiltration of the atrial subepicardium and the electrical disconnection with the rest of the myocardium (PMID: 32175811). This study is not discussed.

Response: This is a very valuable suggestion, thank you. We agree that in addition to potentially changing the phenotypic features of fully differentiated cells such as cardiac myocytes, adipokines or other products of the epicardial adipocyte secretome are likely to alter and direct the cellular differentiation of cardiac resident pluripotent cells to specific phenotypes. The expression of a fibrofatty phenotype from progenitor cells would be markedly maladaptive and pathological. To this effect, we have added this argument and cited reference (PMID: 32175811) in the revised manuscript (Lines 433-442, pages 10 and 11).   

Reviewer 2 Report

This review article suggested that adipose tissue also involved in regulation of myocyte electrophysiology, through the adipokines and/or the electrotonic interactions. I like to suggest the following comments.

  1. Epicardial adipose tissue is important but association with pathologic disorders in addition to obesity seems ignored.
  2. Please move the location of Figure 2 with legends to separate from sentences.
  3. The balance between pro- and anti-inflammatory adipokines during inflammation, fibrosis and excess free oxygen radicals seems not conducted in detail.
  4. The presence of epicardial adipose infiltrates within the myocardium evoked myocardial remodeling which lead to atrial fibrillation that seems the key-point of this disorder. How to link it with adipokines?
  5. The complex interactions in progression of atrial fibrillation are not easy to identify in vitro. How to clarify this problem?
  6. How to prevent the progression of atrial fibrillation? It seems useful.

Author Response

Responses to Reviewer #2

We would like to thank the reviewer for their thoughtful review.

  1. Epicardial adipose tissue is important but association with pathologic disorders in addition to obesity seems ignored.

Response: Thank you for this feedback. As you correctly observe, obesity is not the only disease associated with fibrosis and atrial fibrillation, the key pathological changes we are discussing. However, the wealth of data indicating that epicardial adipose tissue volume increases with increased body mass index, and that both factors are also correlated with an increased risk of developing atrial fibrillation, make obesity an ideal focus to explore the direct and indirect interactions between epicardial adipose tissue, fibroblasts and myocytes in the context of fibrosis and atrial fibrillation. Such correlation is less clear for other cardiovascular diseases which place patients at a higher risk of developing atrial fibrillation (valve disease, chronic atrial dilatation and other structural heart diseases, heart failure, hypertension). For this reason, we described studies associated with atrial fibrillation and obesity preferentially rather than providing a broader but less in-depth over-view of all the pathologies associated with fibrosis and atrial fibrillation.

Interactions between peri-vascular epicardial adipose tissue and cells within the arterial wall are proposed to play a role in the pathogenesis of coronary atherosclerosis (see Antonopoulos and Antoniades 2017, DOI: 10.1113/JP273049, for example). However, such interactions are not presented as directly causing fibrosis and atrial fibrillation, and so, do not represent as good a focus as obesity.

  1. Please move the location of Figure 2 with legends to separate from sentences.

 Response: Thank you for bringing this to our attention. Where appropriate, the figures and table have been repositioned to facilitate ease of reading.

  1. The balance between pro- and anti-inflammatory adipokines during inflammation, fibrosis and excess free oxygen radicals seems not conducted in detail.

 Response:  We agree that this is certainly an important element of the healthy vs. fibrotic, fibrillating atria. We acknowledge this in section 4.2, “Future Directions”, where we emphasize that exploring what causes the switch from healthy to pathological secretomes is an important future direction (page 15, lines 619 to 627). However, we believe that considering the general balance of anti-inflammatory, anti-fibrotic vs. pro-inflammatory and pro-fibrotic adipokines (with increased reactive oxygen species production) is sufficient in investigating modulation of fibroblasts and myocyte electrophysiology by adipose tissue. Furthermore, adipokines have been the subject of numerous previous papers, making an extensive treatment of less value in the current paper. For example, please refer to Mazurek et al. 2003 (DOI: 10.1161/01.CIR.0000099542.57313.C5), Cherian et al., 2012 (10.1152/ajpendo.00061.2012.), Gaborit et al. 2015 (DOI:10.1093/cvr/cvv208) and 2017 (https://hal.archives-ouvertes.fr/hal-01744592), and Smekal and Vaclavik 2017 (DOI: 10.5507/bp.2017.002). 

  1. The presence of epicardial adipose infiltrates within the myocardium evoked myocardial remodeling which lead to atrial fibrillation that seems the key-point of this disorder. How to link it with adipokines?

Response:  Thank you for this astute observation. Intriguingly, fatty infiltrates from the epicardium into the myocardium are observed in healthy myocardium. The literature supports a shift from a fatty to fibrofatty phenotype of these infiltrates in conjunction with the shift in adipokines from protective to pathological, particularly with the development of fibrosis. Discussion of adipokines, fibrosis, and fibrillation with particular attention to the fibrofatty infiltrates has been included in the manuscript within various sections (lines 421 to 432 and 450-454, for example), building an argument that adipose tissue modulates myocardial electrophysiology both directly (by structural and potentially electrotonic means) and indirectly (via adipokines). 

  1. The complex interactions in progression of atrial fibrillation are not easy to identify in vitro. How to clarify this problem?

 Response:  Thank you for the comment. Challenges in understanding atrial fibrillation include the different types of fibrillation, the complex changes over time associated with disease development, and the issues with choosing the correct model. These are identified in the ‘challenges’ section of the manuscript (lines 533 – 534 and 551 – 552). In vitro studies have advantages in offering high levels of control, to ask very specific mechanistic questions such as do adipocytes form functional heterocellular gap junctions, and if so, how would that affect membrane potential? However, their validity is limited, and any conclusions must be drawn with caution. We believe that this caution is included in the description of findings from in vitro studies within the manuscript (for example, noting that 2D or 3D co-culture systems may be required), while still allowing us to speculate within the ‘future directions’ section on how interesting such in vitro experiments would be (lines 606-612).   

  1. How to prevent the progression of atrial fibrillation? It seems useful.

 Response:  This is indeed a useful goal. However, while we have touched on strategies to prevent atrial fibrillation, that was not our primary focus and is beyond the scope of this review. Key insight will come from human studies. For example, ablation studies have considered epicardial adipose tissue while attempting to stop fibrillation from occurring (doi: 10.1016/j.hrthm.2014.04.040, lines 237-239). Studies employing human participants have also shown that weight-loss and control of comorbidities improved the frequency of atrial fibrillation (doi: 10.3389/fphys.2018.00002, lines 239-243). Animal studies considering the development and progression of atrial fibrillation may also shed light on this question. However, as discussed in the section on challenges, the different types of atrial fibrillation may reflect different pathogenesis and progression. This complexity and the expense of conducting long-term studies on mammals suitable to serve as experimental models makes this goal a difficult one to achieve.